# Temporal, Kinematic and Kinetic Variables Derived from a Wearable 3D Inertial Sensor to Estimate Muscle Power during the 5 Sit to Stand Test in Older Individuals: A Validation Study

**DOI:** 10.3390/s23104802

**Published:** 2023-05-16

**Authors:** Gianluca Bochicchio, Luca Ferrari, Alberto Bottari, Francesco Lucertini, Alessandra Scarton, Silvia Pogliaghi

**Affiliations:** 1Department of Neurosciences, Biomedicine and Movement Sciences, University of Verona, 37131 Verona, Italy; gianluca.bochicchio@univr.it (G.B.); luca.ferrari_01@univr.it (L.F.); alberto.bottari@univr.it (A.B.); alessandra.scarton@microgate.it (A.S.); 2Department of Biomolecular Sciences, University of Urbino, 61029 Urbino, Italy; francesco.lucertini@uniurb.it; 3Microgate Srl, 39100 Bolzano, Italy; 4Research Associate Canadian Center for Activity and Ageing, University of Western Ontario, London, ON N6A 3K7, Canada

**Keywords:** IMU, sit-to-stand, validity, ageing, sensors, motion analysis, functional screening

## Abstract

The 5-Sit-to-stand test (5STS) is widely used to estimate lower limb muscle power (MP). An Inertial Measurement Unit (IMU) could be used to obtain objective, accurate and automatic measures of lower limb MP. In 62 older adults (30 F, 66 ± 6 years) we compared (paired *t*-test, Pearson’s correlation coefficient, and Bland-Altman analysis) IMU-based estimates of total trial time (totT), mean concentric time (McT), velocity (McV), force (McF), and MP against laboratory equipment (Lab). While significantly different, Lab vs. IMU measures of totT (8.97 ± 2.44 vs. 8.86 ± 2.45 s, *p* = 0.003), McV (0.35 ± 0.09 vs. 0.27 ± 0.10 m∙s^−1^, *p* < 0.001), McF (673.13 ± 146.43 vs. 653.41 ± 144.58 N, *p* < 0.001) and MP (233.00 ± 70.83 vs. 174.84 ± 71.16 W, *p* < 0.001) had a very large to extremely large correlation (r = 0.99, r = 0.93, and r = 0.97 r = 0.76 and r = 0.79, respectively, for totT, McT, McF, McV and MP). Bland–Altman analysis showed a small, significant bias and good precision for all the variables, but McT. A sensor-based 5STS evaluation appears to be a promising objective and digitalized measure of MP. This approach could offer a practical alternative to the gold standard methods used to measure MP.

## 1. Introduction

Muscle function is defined by muscle strength, muscle power or performance in complex movements (e.g., walking speed) and is positively related to overall health, independence, and quality of life in aging [1]. Among the above indexes, the power of lower limbs is considered the stronger predictor of current and prospective muscle function [2]. Muscle power is lost at a rate of ~3.5% per year after 65 years [3], leading to a progressive loss in independence and mobility which in turn causes an inability to perform the activities of daily living (e.g., recovering balance, walking, sitting and standing from a chair) and further power loss [4]. This vicious cycle increases the risk of adverse health outcomes including falls, hospitalization, institutionalization, and mortality [3,5]. Therefore, it becomes crucial to provide clinicians with accessible tools for assessment and monitoring of the power of the lower limbs in ageing [5].

In health and exercise sciences, the assessment of lower limb muscle power in older adults can be obtained with specific equipment (isokinetic dynamometer, isotonic machines, Nottingham power rig) that requires standardized and relatively unnatural muscle actions [4]. In the laboratory setting, Motion Capture system (MoCap) and force plate are considered the gold standard instruments to measure these variables during a variety of natural movements. However, this equipment is expensive and requires specialized personnel, as well as a time-consuming procedure for data collection and analysis. On the contrary, the ideal approach for testing and monitoring in clinical applications requires affordable costs, a relatively simple and time-efficient procedure, and the use of movements that mimic muscle function in actual daily activities [4].

The 5 sit-to-stand test (5STS test) was developed as a time and cost-efficient field-test for the estimation of power of the lower limbs. Muscle power estimates with 5STS are highly correlated with indexes of functional fitness (e.g., longer time up and go, grip strength, dynamic balance, stair climbing, gait speed) [4,6], frailty [7], and health-related quality of life [8]. While this simplified approach has an undiscussed practical value for testing and periodical monitoring of muscle function, its accuracy and precision may be affected by the accuracy of the time measures (due to manual stopwatch measurement and recording), as well as by the assumptions on which it is based [9]. In this context, body-worn sensors could provide the opportunity to perform objective and digitalized measures of movement and possibly accurate estimates of muscle power. Among body-worn sensors, wearable inertial measurement units (IMU) can be used to record kinematic and kinetic information during a wide range of human movements. Several studies used this technology during the 5 STS test to discriminate between fallers and not fallers [10] or healthy vs. unhealthy individuals during a single sit-to-stand task [11,12,13,14,15,16,17]. While these studies appear promising, a study comparing the validity of the measures of muscle power from the 5STS test based on a 3D inertial sensor (IMU) with a complete gold standard measurement setup (MoCap and force plate) is lacking.

Therefore, the purpose of this study was to assess (1) the accuracy and precision of the IMU-estimated time, velocity, and force and (2) to verify the accuracy and precision of the estimates of lower limb muscle power compared to the gold standard laboratory instruments. Our hypothesis was that IMU could correctly measure duration, velocity, and force. In addition, we hypothesized that lower limb muscle power could be accurately estimated maintaining the ease of use of the field 5STS test but with an objective and automatically digitalized measure.

## 2. Materials and Methods

### 2.1. Study Design

For the purpose of the study, we used a validation design to examine (1) the accuracy and precision of the IMU-estimated time, velocity, and force and (2) to verify the accuracy and precision of the estimates of lower limb muscle power compared to the gold standard laboratory instruments. All participants visited the laboratory, where anthropometric measures were taken and the 5STS test evaluation performed. IMU and laboratory instruments were used simultaneously for the measurement of temporal, kinematic, and kinetic variables during the task.

### 2.2. Participants

A total of 62 independently older adults were recruited by local advertisement (Table 1). The inclusion criterion was age above 60 years. Exclusion criteria were evaluated with a preliminary telephone interview and a successive medical screening to exclude individuals with any orthopedic, mental, or neurological disease that could have interfered with the ability to express lower limb maximal power or a Short Physical Performance Battery (SPPB) score below 9 [18]. All participants signed a written informed consent form prior to participation. All procedures used in the study were approved by the Ethics Committee for Human Research from the University of Verona (28/2023) and conducted in conformity with the Declaration of Helsinki.

### 2.3. Data Collection

During the visit to the laboratory, participant’s anthropometric measures were collected prior to the 5STS test. The anthropometric assessment was performed with participants barefoot and wearing only underwear. Body mass was taken to the nearest 0.1 kg with an electronic scale (Tanita electronic scale BWB-800 MA, Tokyo, Japan) and stature was measured to the nearest 0.01 m with a Harpenden stadiometer (Holtain Ltd., Crymych, Pembs, UK). Body Mass Index (BMI) was calculated as body mass height^−2^ (kg∙m^−2^).

Participants performed the 5STS test for lower limb muscle power determination. Immediately before the test, each participant performed a 10-min warm-up protocol, consisting of 5-min cycling on a cycle-ergometer at a fixed power and cadence (i.e., 50 W at 60 rpm), 4 active mobility exercises for the upper and lower limbs, and 5–6 repetitions of the sit-to-stand movement, which was also considered as a familiarization to the 5STS test. Participants sat on a box (high 0.49 m) with the trunk and shanks positioned perpendicular to the ground and the arms crossed on their chest. The 5STS test consisted of 5 consecutive repetitions of the sit-to-stand movement, executed as quickly as possible. The test was performed twice, with 3-min recovery between the two trials. The trial started after a countdown of three and ended when the participants touched the seat after the fifth repetition [9].

A wearable IMU containing a 3D inertial sensor in combination with a 3D gyroscope and 3D magnetometer (500 Hz, Gyko, Microgate, Bolzano, Italy) was attached to the lateral face of the right thigh of each participant.

A MoCap was used with 8 infrared cameras (100 Hz, Vicon, Oxford, UK) automatically synchronized with a force platform (1000 Hz, AMTI Inc., Watertown, MA, USA) positioned under the participants’ feet. To build a model of the thigh, three markers were placed respectively on the trochanter, mid-thigh (in correspondence with the device), and the lateral epicondyle of the femur.

The IMU and laboratory instruments recorded simultaneously during the trials.

### 2.4. Data Analysis

#### 2.4.1. IMU Data Analysis

Raw data were collected by using the instrument’s dedicated software and subsequently analysed with a self-written MATLAB code. The accelerometer and gyroscope were set to a full scale of 4 g and 2000°∙s^−1^, respectively. The calibration matrix provided by the company was used to convert the raw data in bits to acceleration in m∙s^−2^ and angular velocity in rad∙s^−1^. To reduce the integration error, the initial offset of the gyroscope data was removed. Data from the 3D gyroscope and accelerometer were filtered using a low-pass, second-order Butterworth filter with a 30 Hz cut-off frequency. The cut-off frequency was chosen after a frequency analysis of the signal revealed that there was no significant information above it. The quaternions, which described the orientation of the sensor over time, were computed using the Mahony filter. Euler angles according to the notation ZYX were extracted from the quaternions [19]. The angle around the principal axis of motion was considered for the subsequent analysis (see Figure 1).

The peaks of the Euler angle that corresponded to the maximum rotation of the sensor from the initial position were found. The beginning and the end of the sit-to-stand movement were detected by using a threshold corresponding to 5% of the peaks previously individuated. The duration of all phases was estimated from these data as follows:Sit to Stand transitions = from the threshold at 5% to the peak of the rotation.Stand to Sit transitions = from the peak to the subsequent passage on the threshold of 5%.

#### 2.4.2. Laboratory Instruments Data Analysis

Participants’ kinetic and kinematic variables were measured using the force platform and the MoCap system (on the *z*-axis, perpendicular to the ground). Mid-thigh vertical velocity was automatically extrapolated by the system and used to make all the subsequent computations. Vertical velocity and force signals were low pass filtered at 7 and 15 Hz, respectively, using a second-order Butterworth filter. The identification of the concentric and eccentric phases, as well as the total trial duration, is depicted in Figure 2. Vertical velocity was used to recognize the concentric and eccentric phases, as well as the total trial duration. Positive and negative peaks of vertical velocity were identified and thresholds as 5% of peaks were calculated as follows: (1) the start and end of repetition were found when the vertical velocity reached the positive and negative threshold, respectively; (2) the end of the concentric phase was defined when vertical velocity crossed the zero after positive peak; (3) the start and the end of the total trial were defined as the first positive 5% and the last negative 5%, respectively.

### 2.5. Outcome Measures

#### 2.5.1. IMU Calculations

For the force and velocity estimation, the effect of gravity on the data from the IMU was compensated by rotating the raw acceleration data to align the *z*-axis with the direction of gravity. The magnitude of the acceleration was then calculated and, after that, the following parameters were computed:Time: the total duration of the 5STS test (totT) was computed as the difference between the time coordinates of the first and last 5% thresholds. The duration of the single concentric phases was calculated as the difference between the time coordinates of the positive 5% threshold and the peak of the rotation. Thereafter, the duration of the five single concentric phases was averaged (McT).Velocity: data of the acceleration were segmented by using the instant of beginning and end of the movement previously computed (from the threshold at 5% to the peak of the rotation). These segments were integrated using the Simpson method to obtain the velocity of the movement. The average velocity for each sit-to-stand movement was then extracted (McV).Force: The mean concentric force (McF) was calculated by multiplying the average of the acceleration within each concentric phase with 90% of the body mass of the subject [20].Power: for each repetition, the lower limb muscle power (MP) was computed as the product between mean concentric velocity and mean concentric force.

#### 2.5.2. Laboratory Instruments Calculations

Time: total duration of the 5STS test was computed as the difference between the time coordinates of the first positive and last negative 5% thresholds. The duration of the single concentric phases was calculated as the difference between the time coordinates of the positive 5% threshold and vertical velocity crossing the zero after the positive peak. Thereafter, the duration of the five single concentric phases was averaged.Velocity: velocity was automatically computed from the Vicon. Mean concentric velocity was calculated as the average of the velocity signal within the duration of each concentric phase (Figure 2, top panel). Thereafter, the velocity of the five single concentric phases was averaged.Force: mean concentric force was calculated as the average of the ground reaction force signal within each concentric phase (Figure 2, bottom panel). Thereafter, the force of the five single concentric phases was averaged.Power: for each repetition, lower limb muscle power was computed as the product between mean concentric velocity and mean concentric force.

### 2.6. Statistical Analysis

For the statistical analysis, only the fastest trial for each participant was used. All data were checked for normality using the Shapiro–Wilk test. An unpaired *t*-test was run to compare anthropometric variables between females and males. Paired *t*-test, Pearson correlation coefficient, and Bland-Altman analysis were run to test differences and absolute level of agreement between IMU estimates and laboratory measures of totT, McT, McV, McF, and MP. The correlation coefficient was interpreted according to the values of the r: trivial (<0.1); small (0.10–0.29); moderate (0.30–0.49); large (0.50–0.69); very large (0.70–0.89); extremely large (0.90–1.00) [21].

The Bland–Altman analysis was followed by a one-sided z-test on the bias. Bland–Altman analysis [22] was used to determine potential systematic bias, reporting mean bias, limits of agreement (LOA), and coefficient of determination (R^2^) from regression analysis between differences and means of IMU and laboratory measures of power. Data are reported as mean ± SD. The level of significance was set at 0.05. The SigmaPlot 12.0 software (SigmaStat, San Jose, CA, USA) was used to conduct all the statistical analyses.

## 3. Results

All the participants were able to perform the whole procedure properly, it was well tolerated and no adverse events were recorded. In addition, no subjects or trials have been discarded.

Anagraphics, anthropometric measures and SPPB scores of the recruited subjects are reported in Table 1. Duration, velocity, force, and power variables of all the participants are reported in Table 2.

Paired *t*-test showed a significant difference for all the variables (*p* < 0.05) except mean concentric time (*p* = 0.890). Pearson correlation coefficient (Figure 3, left panel) showed an extremely large correlation for the total time trial, mean concentric time, and mean concentric force (r = 0.99, r = 0.93, and r = 0.97, respectively) and a very large correlation for mean concentric velocity (r = 0.76), and mean concentric power (r = 0.79). Bland–Altman analysis showed a significant bias (for all parameters but mean concentric time, which displayed a non-significant bias) and good precision for all the variables (Figure 3, right panel).

## 4. Discussion

This is the first study that tested the accuracy of an IMU in estimating muscle power during the 5STS test in comparison with a fully objective, gold standard and automated method. Our data indicate that a single IMU placed on the lateral face of the thigh provides estimates of kinetic and kinematic indexes of muscle action, as well as of muscle power, during the 5STS test, which are highly correlated with gold standard laboratory measures. While a significant difference was recorded between measures, likely due to the sensor placement, IMU appears to offer a promising practical alternative to the gold standard methods used to measure MP.

The present results showed that an automated analysis of the instrumented 5STS using a 3D inertial sensor is feasible. Indeed, none of the 62 trials have been discarded due to signal problems and our algorithm was able to correctly identify each phase (i.e., concentric and eccentric phase) of the 5STS and successfully extract time, velocity, and force variables from the 3D inertial sensor signals of all trials.

### 4.1. Time

The values of Total time from the laboratory approach (8.97 ± 2.44 s) and IMU (8.86 ± 2.45 s) obtained in our study are comparable to those found in the literature (from ~8 s to ~15 s, for not fallers and fallers of both sexes, 65–90 years old) [7,9,10,16,23,24]. In our study, we found a value of IMU-based Total time that differs from the time measured with the laboratory instruments (*p* = 0.003). While statistically significant, the absolute difference is ~0.12 s (i.e., 1.2% of the average measure), which is less than the minimum detectable change for the 5STS test [25]. As such, this difference could be interpreted as not clinically or practically relevant. In addition, we found an almost perfect correlation (r = 0.99) between the total time of IMU and laboratory measures. Finally, the bias and the limits of agreement (bias = −0.11 s, z-score = −3.07, LOA= (0.43–0.64 s)) were lower than the values found in other studies (bias = 0.48 s, LOA = 0.32 s) [16] and a null relationship was observed (R^2^ = 0.003), indicating that the difference between methods is similar across the entire range of time measures. Therefore, we can conclude that IMU is a valid approach to measurement of the total time of a 5STS test.

Instrumenting the 5STS test could be a cornerstone in muscle function assessment, because the instrumentation would provide more information than a simple chronometer used for collecting the time to complete the task. Indeed, a wearable device could discriminate between each repetition and, within them, each standing or sitting phase, returning, for example, the time effect of muscle fatigue or variability within the task [26,27]. In our study, we found a lower value of mean concentric time for laboratory instruments (0.56 ± 0.15 s) compared to the literature (range from ~0.8 to ~1.5 s) [28,29,30]. Conversely, greater mean concentric time values were found by the IMU (0.56 ± 0.16 s) compared to the literature (0.41 ± 0.20 s) [31]. These discrepancies may be due to the different methods used for discriminating the standing phase, to the different types of population studied (healthy older subjects vs. random community-dwelling, individuals affected by stroke) or to the different heights of the chair (in our case fixed at 49 cm vs. adjusted according to participants’ anthropometry).

Within our study, IMU-based measures of mean concentric time were not different, very highly correlated (r = 0.93), without significant bias (bias = 0.00, z-score =0.14), and with small limits of agreement (L.O.A.= [0.12 s–0.12 s]) compared to the gold-standard laboratory approach. This would indicate that the method used for the data analysis of the IMU signal identifies the same “temporal windows” as the laboratory approach. Since we calculated the values of force and velocity within these temporal windows, we can conclude that the differences in these variables are not due to the difference in phase discrimination or instrument synchronization. Therefore, the IMU is a valid method to measure accurately and precisely the mean concentric time and these values can be used during clinical assessments.

### 4.2. Velocity

In our study, we found values of mean concentric velocity that are lower compared with the values of other studies for both laboratory equipment (0.35 ± 0.09 m·s^−1^ vs. ~0.50 m·s^−1^) [17,24,32] and IMU (0.27 ± 0.10 m·s^−1^ vs. ~0.65 m·s^−1^) [17,33]. This difference cannot be explained by age, BMI or low function indexes [34], since our population sample is younger compared to the references (66.7 ± 5.9 years vs. 71 to 78 years), with similar BMI and characterized by high functionality, as indicated by the average SPPB score (Table 1) [24,33]. Therefore, this discrepancy may have lain in different methods used to calculate the velocity (Video, MoCap or force plate). Indeed, the most likely source of discrepancy may be that we calculated the mid-thigh velocity instead of the center of mass velocity (i.e., trochanter level) or trunk velocity. Despite that angular velocity is the same during a rotation of a rigid segment independently of the distance from the fulcrum, the tangential velocity is directly influenced by the radius (v = ω*radius). Since we calculated the velocity at the mid-thigh (nearer to the knee, i.e., the fulcrum of the thigh rotation), we found lower values of linear velocity compared to the values calculated in the literature. Indeed, if we recalculate the linear velocity considering laboratory equipment and the trochanter marker rather than the mid-thing placing, the values align with those found in previous studies (0.49 ± 0.10 m·s^−1^).

In our study, we found that the mean concentric velocity calculated by the IMU (0.27 ± 0.10 m·s^−1^) was lower (*p* < 0.001) than the values calculated by the laboratory equipment (0.35 ± 0.09 m·s^−1^). This discrepancy could lie in three methodological factors:The procedure of the phase’s identifications. To discriminate each phase and repetition of the 5STS test from the IMU and the laboratory equipment, we considered the Euler angle and the marker’s vertical velocity signal, respectively. The same method for phase identification was applied to both signals (see method section). Therefore, the differences observed in mean concentric velocity could lie in the fact that we applied the same methodological approach to two distinct types of signal.Another source of discrepancy may lie in the process used to compute the velocity from the IMU raw data. Indeed, the velocity was calculated by numerical integration of the acceleration, where zero values were imposed as integration limits, considering as null the velocity at the beginning and the end of the standing motion. However, since we used the time boundaries from the Euler angle signal (as shown in Figure 1), the velocity in correspondence with these limits during the standing movement was not exactly zero. This could lead to an underestimation of mean concentric velocity.Amplitude of the sensor-based acceleration signal. Multiple studies in the literature tried to estimate the ground reaction force by using accelerometer data. They discovered that, the closer the IMU is to the fulcrum, the smaller the amplitude of the acceleration signal measured [35]. Therefore, the mid-thigh IMU may have been subjected to lower absolute values of acceleration.

Overall, our results indicate that, while IMU slightly underestimates the velocity of the movement compared to gold standard methods, possibly due to methodological factors, it displays a very high correlation (r = 0.76), and significant and constant bias (bias= −0.08 m·s^−1^, z-score= −9.85, R^2^ = 0.030) with small limits of agreement (L.O.A. = (0.05–0.20 m·s^−1^)). Thus, the IMU could be considered overall a valid and precise instrument for the estimation and monitoring of mean concentric velocity.

### 4.3. Force

In this study, the mean concentric force values for both methods (~510–790 N) were similar to the values found in the literature (~550–700 N) [9,24,36,37,38,39]. Regarding the comparison between methods, we found that the force values were different (*p* < 0.001) but with almost perfect correlation (r = 0.97), with a very small significant bias (bias = −20 N; z-score = −4.78), and small limits of agreement (L.O.A. = [44–84 N]). It is worth mentioning that the difference between methods corresponds to only ~2 kg and therefore could be considered as not practically relevant. Previous studies found that, during the sit-to-stand task, not all of the body mass of the subject is accelerated, and different percentages of the body mass were found (90%, 87%, and 67% of body weight) [9,20,24]. The differences in these values could be due to different equipment, methods of phase discrimination, and assumptions used to identify the concentric phase. In this study, we considered that only 90% of body mass was accelerated and, therefore, for the IMU, we calculated the mean force by multiplying this value by the mean concentric acceleration. Furthermore, most of the studies use more than one IMU sensor and develop models to describe the body as a linked chain of multiple elements in order to correctly estimate the force [36]. The use of one single sensor on the thigh could thus have determined a systematic underestimation of the absolute value of force.

### 4.4. Power

Peak values (~700–900 W) [28,40] or mean values (~300–600 W) [2,32] of the 5STS assessed with different instruments (wearable devices, force plate, motion capture, stopwatch) found in the literature are greater than our results (laboratory equipment = peak 414.24 ± 117.01 W and mean 233.00 ± 70.83 W; IMU = peak 348.23 ± 114.21 and mean 174.84 ± 71.16 W). Since power was computed as the mechanical product between force and velocity, and considering that force was similar to the literature’s values, power differences are attributable to our lower mean concentric velocity measured at the mid-thigh. To corroborate this view, if we recalculate the power considering laboratory equipment and the trochanter marker rather than the mid-thing placing, the values align better with those found in previous studies (peak power 556.93 ± 168.83 W 295.20 ± 89.78 W; mean power 295.16 ± 98.09 W).

In addition, IMU power values were significantly lower than the values measured with laboratory equipment (*p* < 0.001). As previously discussed for the velocity signal, this discrepancy could be attributable to: (i) phase identification procedure, (ii) velocity calculation process and (iii) effect of sensor placement on acceleration. While the bias was significantly different, the limits of agreement were relatively small (bias = −58 W, z-score = −10.07, LOA = (32–150 W)) and a null relationship was observed (R^2^ = 0.000). The above indicates a good repeatability and a systematic bias, i.e., that the difference between instruments is similar across the entire range of power.

### 4.5. Limitations and Future Developments

The present study has some limitations. We placed the IMU on the lateral face of the mid-thigh, and this could have led to lower values of velocity and power compared to the literature. The most common sensors/markers placements vary from the front sternum or chest (27%) to the back trunk (57%) (i.e., L5–L3) or on the lower limb (44.5%) (such as the thigh, shank or ankle) [27]. We chose the mid-thigh, among others, because it is more accessible and facilitates the placing of the IMU even in the presence of a weight vest (in view of a possible future application of the method). The back trunk positioning was excluded because we wanted to isolate the sole leg muscle power by avoiding the possible noise from the oscillations of the trunk [13]. While the sensor placement does not jeopardise the comparison between methods, which is the focus of our study, it is plausible that the muscle power measured at the middle-thigh underestimates the lower limb’s ability to express power. Therefore, a mathematical model for estimating the whole system power (i.e., centre of mass or trochanter) by wearing the IMU on the lateral face of the mid-thigh could be considered for future studies.

Another limitation is that we used the same chair (chair height = 0.49 m) for all the participants and, therefore, we did not control for the individual articular angle/muscle length. Indeed, a fixed-height chair leads to different articular angles/muscle lengths for people with different leg length; in turn, this may lead to a different expression of force over time [41]. While this is unlikely to affect the correspondence between methods of measurement, the standardisation of knee angle between subjects will facilitate the interpretation of data across individuals with different anthropometric characteristics (i.e., short vs. long lower limb length).

## 5. Conclusions

The 5STS test is a valuable and common test used to estimate lower limb muscle power, thanks to its simplicity and low cost. However, in its field version (i.e., human eye and stopwatch), it is not independent of the operator’s error. In our study, we found a not significant bias between measures of kinetic and kinematic parameters and muscle power measured with either a wearable IMU or a full, gold-standard laboratory approach. Therefore, a wearable sensor-based 5STS evaluation could allow a valid, low-cost alternative to the gold standard methods classically used to measure muscle power; moreover, it could be a more repeatable, objective, and immediately digitalized option compared to the stopwatch method.

## Figures and Tables

**Figure 1 sensors-23-04802-f001:**
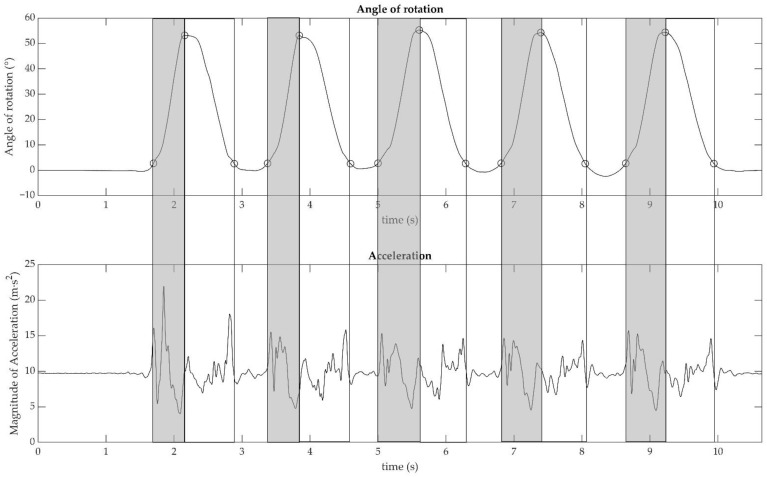
Angle of rotation around the main axis (expressed in degree °, upper graphic) and Magnitude of acceleration (m·s^−2^, lower graphic) are plotted as a function of time during the 5 Sit to Stand test in a representative subject. The points (○) mark the events of each repetition (start, standing position, and end). Light grey sections highlight the concentric phases (i.e., raising phase), while white sections highlight the eccentric phases (i.e., sitting phase).

**Figure 2 sensors-23-04802-f002:**
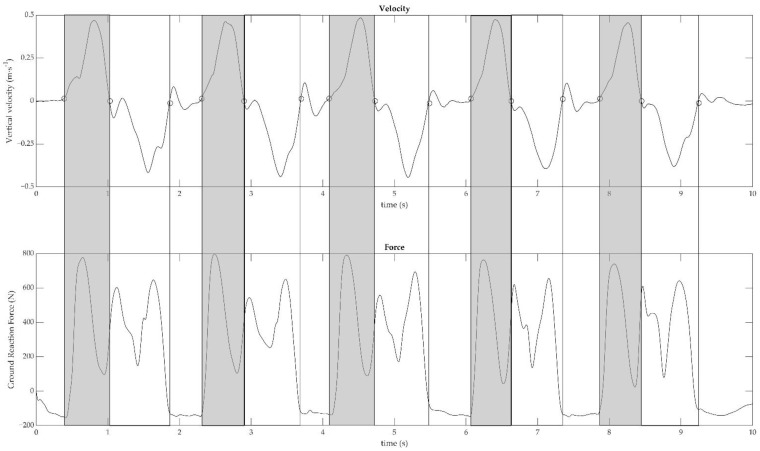
Vertical velocity (m s^−1^; above) measured by motion capture and vertical force (N; below) measured by force plate in the 5 Sit to Stand test are plotted in function on time (s) in a representative participant. The points (○) mark the events of each repetition (start, standing position, and end). Light grey sections highlight the concentric phases (i.e., raising phase) while white sections highlight the eccentric phases (i.e., sitting phase).

**Figure 3 sensors-23-04802-f003:**
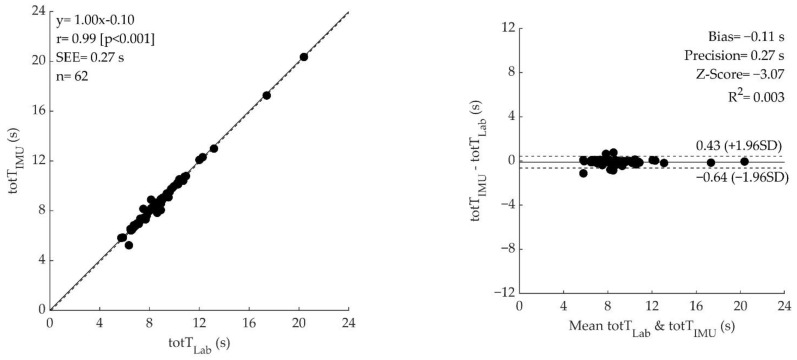
The left side of the figure shows the correlation plots between laboratory and IMU measures of 5 sit-to-stand total time (totT), mean concentric time (McT), mean concentric velocity (McV), mean concentric force (McF) and mean power (MP). Equation, Pearson’s correlation coefficient (r), *p*-value, SEE and sample size are reported along with regression (dashed) and identity (solid) lines. The right side of the figure shows the Bland Altman analysis comparing laboratory and IMU of the same variables. Individual differences between lab and IMU measures are plotted as a function of the mean of the two. Bias, R^2^, and Z-score are reported along with the LOA (dashed lines) and bias (solid lines).

**Table 1 sensors-23-04802-t001:** Anthropometric characteristics of the participants.

	**Females**	**Males**	**Tot**	** *p-* ** **Value**
**#**	**30**	**32**	**62**	
Age (yrs)	65.3 ± 5.2	68.0 ± 6.3	66.7 ± 5.9	0.072
Weight (kg)	65.5 ± 11.4	81.8 ± 14.6	73.9 ± 15.4	<0.001
Height (m)	1.61 ± 0.06	1.74 ± 0.07	1.68 ± 0.09	<0.001
BMI (kg∙m^−2^)	25.2 ± 4.1	27.1 ± 5.0	26.2 ± 4.6	0.112
SPPB score	11.0 ± 0.9	11.0 ± 0.7	11.0 ± 0.8	0.580

BMI, Body Mass Index; SPPB score, short physical performance battery score. Significant differences between sexes are indicated with the *p*-value < 0.05 (unpaired *t*-test on means).

**Table 2 sensors-23-04802-t002:** Kinetic and kinematic variables of all participants measured with Lab and IMU methods.

		Lab	IMU	*p*-Value
**#**		**62**	**62**	
Duration	totT (s)	8.97 ± 2.44	8.86 ± 2.45	0.003
	McT (s)	0.56 ± 0.15	0.56 ± 0.16	0.890
Velocity	McV (m∙s^−1^)	0.35 ± 0.09	0.27 ± 0.10	<0.001
Force	McF (N)	673.13 ± 146.43	653.41 ± 144.58	<0.001
Power	MP (W)	233.00 ± 70.83	174.84 ± 71.16	<0.001

Lab= laboratory method; IMU= 3D inertial sensor method; totT= total time; McT= mean concentric time; McV= mean concentric velocity; McF= mean concentric force; MP = mean concentric power. Significant differences are indicated with the *p*-value < 0.05 (paired *t*-test on means).

## Data Availability

The data presented in this study are available on request from the corresponding author. The data are not publicly available due to restrictions (privacy).

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
