# Peer review of "Temporal, Kinematic and Kinetic Variables Derived from a Wearable 3D Inertial Sensor to Estimate Muscle Power during the 5 Sit to Stand Test in Older Individuals: A Validation Study"

_sensors, 2023, doi:10.3390/s23104802_

Round 1

Reviewer 1 Report

General comments: the study presented is an attempt to validate a device for assessing muscle performance during the 5 STS test. The population sample is significant. Conclusions should be more cautious: despite significant correlations and adequate agreements, there is a significant discrepancy between IMU sensor estimates and lab results. 

Specifics comments: 

Procedures: it seems 5 STS test was performed twice by all participants, but the sequence of muscle work evaluation methods is not clear: were both methods used simultaneously or was a single method applied for each  5 STS test?  

Results: it would be interesting to know if all participants were able to perform the whole procedure properly and if it was well tolerated. It is also recommended to mention if participants were excluded from de study and for what reason. Number of tests analyzed should be added Table 2. 

Discussion: feasability of the test should be emphasized considering the significant number of participants. The variability of lower limbs muscle mass was not really taken into account. This point should come up in the discussion. Apparent underestimation of muscle work by the IMU sensor should be further discussed. A more systematic description of potential biases would probably improve the discussion section.

Reviewer 2 Report

Although a clinical assessment has been made not to exclude different pathologies in the older adults sample, it would have been interesting to understand functional characteristics using questionnaires related to physical activity and lower extremity function (i.e: Low Extremity Biomechanical Index and IPAQ). In that way, applying mixed models may help to understand the effect of age, BMI or other variables as explanation of possible differences of the sample compared to the references that appear in the discussion.

In the methods, of comparison with a gold standard, it would have been interesting to use a complete modeling, at least of the hip, to allow the obtaining of .moments and powers extracted from the platformd e forces. An EMG record could have been useful for obtaining a muscle signal. 

Reviewer 4 Report

The Authors focused on the study of Temporal, kinematic and kinetic variables derived from a wearable 3D inertial sensor to estimate muscle power during the 5 Sit 3 to stand test in older individuals: a validation study. 

Overall, it is a well-written and comprehensive article, with clear tables and figures, interesting results that can make an important contribution to further large studies. 

In my opinion:

- The abstract presents an accurate description of this study.

- The authors have performed an adequate literature review.

- References support the rationale for reporting the study.

- The subjects are described adequately.

- The management of the study is effectively described.

- Valid and reliable outcome measures were used.

- Conclusions are appropriate and comprehensive.

Reviewer 5 Report

The manuscript by Bochicchio et al, aimed to validate a wearable sensor for a very common test, 5 STS. The use seems appropriate to me (automatic method to quantify the performance on the 5 times sit-to-stand test) and the validation adequate.

 I was wondering if the authors recorded some gender differences.

I suggest also to correct some typos.
